# Matrix product operator representations for the local conserved quantities of the Heisenberg chain

## Kyoichi Yamada[*] and Kohei Fukai[†]

The Institute for Solid State Physics, The University of Tokyo,
Kashiwa, Chiba 277-8581, Japan

[*] kyamada.phys@gmail.com , [†] k.fukai@issp.u-tokyo.ac.jp

## Abstract

We present the explicit expressions for the matrix product operator (MPO) representation for the local conserved quantities of the Heisenberg chain. The bond dimension of the MPO grows linearly with the locality of the charges. The MPO has more simple form than the local charges themselves, and their Catalan tree patterns naturally emerge from the matrix products. The MPO representation of local conserved quantities is generalized to the integrable SU($N$) invariant spin chain.

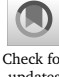
## 1   Introduction

Quantum integrable models are special many-body systems that allow for exact solutions [1,2]. The Bethe Ansatz, with its origins tracing back to Hans Bethe's seminal work on the exact solution of the spin-1/2 Heisenberg spin chain [3], enables the exact calculation of energy spectrum and physical observables. Through various generalizations, the Bethe Ansatz has become the most renowned method for solving integrable models.

The defining feature of quantum integrable systems is the presence of an extensive number of local conserved quantities, denoted as $\{Q_k\}_{k=2,3,4,\ldots}$. They are local in the sense that they are a linear combination of operators that act on a finite range of local sites. In our notation here, $Q_k$ is the site translation sum of the local operator that acts on the adjacent $k$ sites. The existence of these local conserved quantities has been well established by the quantum inverse scattering method [4]: the local conserved quantities can be derived from the expansion of the transfer matrix $T(\lambda)$ with respect to the spectral parameter $\lambda$, given by $\log T(\lambda) \sim \sum_{k \geq 2} \lambda^{k-1} Q_k$, where $Q_2$ is usually the Hamiltonian itself. The commutativity of the transfer matrix, $[T(\lambda), T(\mu)] = 0$, ensures the mutual commutativity of the local conserved quantities, $[Q_k, Q_l] = 0$. Another way to obtain $Q_k$ is the usage of the Boost operator, denoted by $B$, if it exists. $Q_k$ can be calculated recursively by $[Q_k, B] = Q_{k+1}$.

Although the formal methodology for generating local conserved quantities $Q_k$, through the expansion of the transfer matrix and using the Boost operator, has been known, determining their general expressions in practice remains a formidable challenge. This difficulty stems not only from the excessive computational expense associated with higher-order charges but also from finding a general pattern within the vast data sets generated by these calculations. The general forms of local conserved quantities were firstly found for the spin-1/2 Heisenberg chain(XXX chain) independently by [5] and [6] and subsequently found for its SU($N$) generalization [7]. For these models, the structure of the local charges has the Catalan tree pattern [8]: $Q_k$ is constructed from the linear combination of the polynomial of spin operators with the coefficient of generalized Catalan number. More recently, the general forms of the local charges were found in the spin-1/2 XYZ chain [9], the Temperly-Lieb models [10], which include the spin-1/2 XXZ chain, and the one-dimensional Hubbard model [11]. Nonetheless, their expressions are still slightly complicated, even for these models where the general forms of the local charges are now known. A universal, more simple description of the general form of local conserved quantities is highly desirable.

Over the past two decades, there has been a growing trend to introduce tensor network techniques [12–14] to reformulate quantum integrability. The matrix product state(MPS) representation of the Bethe eigenstate of quantum integrable systems has been studied [15–22]. The symmetry of integrable systems was also investigated in terms of matrix product operator(MPO). In [23], the MPO commuting with the Hamiltonian of the Heisenberg chain was constructed, which was proved to be the product of two transfer matrices with different spectral parameters. More recently, the non-commutative symmetry of models with fragmented Hilbert space was obtained in the form of MPO, even for non-integrable cases [24]. The hidden

symmetry of an integrable Lindblad system, which leads to multiple non-equilibrium steady states, was exactly given by MPO [25]. However, despite the fact that the transfer matrix, which is the source of $Q_k$, is defined by the MPO constructed from the Lax operator, there has yet to be an exploration of the MPO representation for local conserved quantities themselves. For example, the transfer matrix of the SU(N) invariant chain is the MPO with bond dimension $N$. It is worth noting that the fact that the transfer matrix is expressed as an MPO had been known even before the term "MPO" was coined.

In this work, we present the MPO representation of the local conserved quantities for the spin-1/2 Heisenberg spin chains. The local conserved quantity $Q_k$ can be represented by the MPO whose bond dimension is $3k-1$. We found the Catalan tree pattern of the local conserved quantities [8] naturally emerges from the product of the MPO by using the identity of the generalized Catalan number. Unlike the local conserved quantities themselves, their MPO representation is only involved with the usual Catalan number, implying that the complexity of these expressions is folded within the product of the MPO. This MPO representation of the local conserved quantities can be immediately generalized for the integrable SU($N$) invariant spin chains of fundamental representation. To our knowledge, this is the first study investigating the MPO representation for local conserved quantities of quantum integrable systems.

This paper consists of the following Sections: in Section 2, we review the local charges for the Heisenberg chain and the SU($N$) integrable spin chain. In Section 3, we present the main result of the MPO representation for the local charges in the Heisenberg chain. In section 4, we give the demonstration that the MPO introduced in section 3 actually reproduces the local charges of [5–7]. Section 5 contains the summary of our work and future outlooks.

## 2 Local conserved quantities of the Heisenberg chain

In this section, we review the result of the local conserved quantities of the Heisenberg chain [5,6] and its SU($N$) generalization [7].

### 2.1 Spin-1/2 Heisenberg chain

The Hamiltonian of the spin-1/2 Heisenberg chain is given by:

$$H = \sum_{i=1}^{L} \boldsymbol{\sigma}_i \cdot \boldsymbol{\sigma}_{i+1}, \tag{1}$$

where $\boldsymbol{\sigma}_i = (X_i, Y_i, Z_i)$ stands for the vector of the usual Pauli matrices acting non-trivially on $i$-th site, and $L$ is the system size. We assume the periodic boundary condition: $\boldsymbol{\sigma}_{i+L} = \boldsymbol{\sigma}_i$. The Hamiltonian (1) is integrable [3] and has an extensive number of local conserved quantities $\{Q_k\}_{k=2,3,4,\dots}$. The key sign of its integrability is the mutual commutativity $[Q_k, Q_l] = 0$, and $Q_2 = H$ is the Hamiltonian itself.

To represent the expression of $Q_k$, we introduce some notations. A sequence of $n$ sites $\mathcal{C} = \{i_1, i_2, \dots, i_n\}$ with $i_1 < i_2 < \dots < i_n$, will be called a *cluster* of order $n$. A cluster $\mathcal{C}$ can be further classified by *hole*, defined by $i_n - i_1 + 1 - n$, which is the number of the sites between $i_1$ and $i_n$ that are not included in $\mathcal{C}$. For example, the cluster $\mathcal{C} = \{1, 3, 4, 7\}$ is the cluster of order 4, and whose hole is 3.

For a cluster $\mathcal{C} = \{i_1, i_2, \dots, i_n\}$ of order $n$, we define the nested products of Pauli matrices $f_n(\mathcal{C})$ by:

$$f_n(\mathcal{C}) := \boldsymbol{\sigma}_{i_1} \cdot \left( \boldsymbol{\sigma}_{i_2} \times \left( \boldsymbol{\sigma}_{i_3} \times \left( \cdots \times \left( \boldsymbol{\sigma}_{i_{n-1}} \times \boldsymbol{\sigma}_{i_n} \right) \cdots \right) \right) \right). \tag{2}$$

Then we define components of the local conserved quantities:

$$F_{n,m} = \sum_{\mathcal{C} \in \mathcal{C}^{(n,m)}} f_n(\mathcal{C}), \tag{3}$$

where $\mathcal{C}^{(n,m)}$ denotes the set of clusters of order $n$ with $m$ holes, satisfying $1 \le i_1 \le L$.

The general expression of $Q_k$ is given by [5, 6]:

$$Q_k = F_{k,0} + \sum_{\substack{1 \le n+m < \lfloor k/2 \rfloor \\ 0 \le n, 1 \le m}} C_{n+m-1,n} F_{k-2(n+m),m}, \tag{4}$$

where $C_{k,n} \equiv \binom{k+n}{n} - \binom{k+n}{n-1} = \frac{k+1-n}{k+1}\binom{k+n}{n}$ is the generalized Catalan number.[1]

## 2.2 SU($N$) generalization

The structure of the local conserved quantities of the isotropic SU($N$) version of the spin-1/2 Heisenberg chain in the fundamental representation is the same as that of the spin-1/2 Heisenberg chain [7]. The Hamiltonian of SU($N$) invariant chain is given by [26]:

$$H = \sum_{i=1}^{L} \sum_{a=1}^{N^2-1} t_i^a t_{i+1}^a, \tag{5}$$

where $t_i^a, a = 1, \ldots, N^2 - 1$ are the $su(N)$ generators in the fundamental representation. For the $N = 2$ case, (5) reduces to the Hamiltonian of the spin-1/2 Heisenberg chain. We choose the normalization of the generators so that $t_a$'s are the $su(N)$ Gell-Mann matrix, satisfying the following algebra:

$$\left[t^a, t^b\right] = 2i f^{abc} t^c, \tag{6}$$

$$t^a t^b + t^b t^a = \frac{4}{N} \delta_{ab} + 2 d^{abc} t^c, \tag{7}$$

where $f^{abc}$ is the structure constant of $su(N)$, and $d^{abc}$ is a completely symmetric tensor, which is non-trivial for $N > 2$.

For the SU($N$) case, $f_n(\mathcal{C})$ for a cluster $\mathcal{C} = \{i_1, i_2, \ldots, i_n\}$ is defined by:

$$f_n(\mathcal{C}) := \mathbf{t}_{i_1} \cdot \left(\mathbf{t}_{i_2} \times \left(\mathbf{t}_{i_3} \times \left(\cdots \times \left(\mathbf{t}_{i_{n-1}} \times \mathbf{t}_{i_n}\right)\cdots\right)\right)\right), \tag{8}$$

where $\mathbf{t}_i = \left(t_i^1, \ldots, t_i^{N^2-1}\right)$ stands for the vector of the $su(N)$ Gell-Mann matrices acting non-trivially on $i$-th site. The outer product of the vector $\mathbf{A}, \mathbf{B}$ with $N^2 - 1$ elements is defined by $(\mathbf{A} \times \mathbf{B})^c \equiv f^{abc} A^a B^b$.

The local conserved quantities of the SU($N$) invariant chain are expressed in the same form as (4), with $f_n(\mathcal{C})$ defined in (8).

---

[1]The coefficients of local charges of Eq. (4.2) in [7] are slightly complicated; thus, we used more simple notation here. The relation between our $C_{k,n}$ and $\alpha_{k,l}$ of Eq. (4.2) in [7] is $C_{k,n} = \alpha_{k+1,k+1-n}$. This can be easily seen by using Eq. (4.5) in [7]: $\alpha_{k+1,k+1-n} = \binom{k+n-1}{n} - \binom{k+n-1}{n-2} = \left(\binom{k+n}{n} - \binom{k+n-1}{n-1}\right) - \left(\binom{k+n}{n-1} - \binom{k+n-1}{n-1}\right) = \binom{k+n}{n} - \binom{k+n}{n-1} = C_{k,n}$. Note that the definition of the symbol $C_{n,m}$ employed in [7] is distinct from the generalized Catalan number in this work.

## 2.3 Doubling-product representation for SU(2) case

For the SU(2) case, the local conserved quantities can be represented using the *doubling-product* notation, which was initially introduced in the proof of the non-integrability of the spin-1/2 XYZ chain in a magnetic field [27], and subsequently employed to construct the local conserved quantities of the spin-1/2 XYZ chain without a magnetic field [9].

A doubling-product is a notation for the product of the Pauli matrices, defined by:

$$\overline{A_1 A_2 \cdots A_n} := \sum_{i=1}^{L} (A_1)_i (A_1 A_2)_{i+1} (A_2 A_3)_{i+2} \cdots (A_{n-1} A_n)_{i+n-1} (A_n)_{i+n}, \tag{9}$$

where $A_i \in \{X, Y, Z\}$ and $A_l A_{l+1}$ is the product of $A_l$ and $A_{l+1}$. $(\cdot)_i$ denotes the operator acting on the $i$-th site. The hole is equal to the number of $j$ that satisfies $A_j = A_{j+1}$. We define the support of an operator as the range of sites on which it acts. The support of the doubling-product of (9) is $n + 1$.

With the doubling-product notation, components of local conserved quantities for the spin-1/2 Heisenberg chain (3) can be rewritten by:

$$F_{n,m} = i^n \sum_{\overline{A} \in \mathcal{S}_{n+m,m}} \overline{A}, \tag{10}$$

where $\mathcal{S}_{l,m}$ is the set of all doubling-products with (support, hole) $= (l, m)$. The local conserved quantities constructed with (10) differ from those constructed with (3) by the factor $i^n$.

# 3 MPO representation for the local conserved quantities

In this section, we present the matrix product operator (MPO) representation for the local conserved quantities of the spin-1/2 Heisenberg chain and its SU($N$) generalizations. Given an operator as a sum of finite-range interactions, it is possible to construct the MPO representation for such a local operator using entirely upper (or lower) triangular matrices [28]. We show the local conserved quantities can be represented by an open boundary MPO constructed with the upper triangular matrix $\Gamma_k^i$.

## 3.1 MPO for the Hamiltonian

MPO component for the Hamiltonian of the spin-1/2 Heisenberg chain (1) has been known as the matrix with bond dimension 5, the element of which are local operators acting on the physical Hilbert space [29]:

$$\Gamma_2^i := \begin{pmatrix} I & X_i & Y_i & Z_i & O \\ O & O & O & O & X_i \\ O & O & O & O & Y_i \\ O & O & O & O & Z_i \\ O & O & O & O & I \end{pmatrix} = \begin{pmatrix} I & \boldsymbol{\sigma}_i & O \\ & O_{4,4} & \boldsymbol{\sigma}_i^\top \\ & & I \end{pmatrix}, \tag{11}$$

where $\boldsymbol{\sigma}_i = (X_i, Y_i, Z_i)$ is treated as a row vector and $\boldsymbol{\sigma}_i^\top$ is its transpose and $I$ is the identity operator. $O$ denotes the zero-operator, and $O_{m,n}$ is an $m \times n$ matrix whose entries are all $O$.

The Hamiltonian is reproduced by the MPO constructed from $\Gamma_2^i$:

$$\Gamma_2^1 \Gamma_2^2 \cdots \Gamma_2^L = \begin{pmatrix} I & \boldsymbol{\sigma}_L & H^c \\ & O_{4,4} & \boldsymbol{\sigma}_1^\top \\ & & I \end{pmatrix}, \tag{12}$$

where (1, 5)-component of the RHS, $H^c = \sum_{i=1}^{L-1} \boldsymbol{\sigma}_i \cdot \boldsymbol{\sigma}_{i+1}$, is the bulk term of the Hamiltonian, that misses the boundary term, $h^B = \boldsymbol{\sigma}_L \cdot \boldsymbol{\sigma}_1 = \boldsymbol{\sigma}_L \boldsymbol{\sigma}_1^\top$. The bulk term can be written simply with the boundary vectors $\langle L | = (1, 0, 0, 0, 0)$ and $|R\rangle = (0, 0, 0, 0, 1)^\top$: $H^c = \langle L | \Gamma_2^1 \Gamma_2^2 \cdots \Gamma_2^L | R \rangle$. The Hamiltonian under the periodic boundary condition (1) is reproduced by the sum of the bulk term and boundary term: $H = H^c + h^B$. In the following, we generalize this result to the higher-order local conserved quantities $Q_k$.

## 3.2 Building blocks of MPO

We introduce a $3 \times 3$ matrix $M_i$, components of MPO which are local operators acting on the physical Hilbert space. $M_i$ serves as the building block of the MPO for the local conserved quantities of the spin-1/2 Heisenberg chain. $M_i$ is defined by:

$$M_i := \begin{pmatrix} O & -Z_i & Y_i \\ Z_i & O & -X_i \\ -Y_i & X_i & O \end{pmatrix}, \tag{13}$$

where the off-diagonal elements are the Pauli matrices acting on the $i$-th site. Note that $M_i$ has the form of the $so(3)$ generator assigned with the Pauli matrices. The nested product of (2) is represented with these building blocks by:

$$\begin{aligned} f_n(i_1, i_2, \ldots, i_n) &= \boldsymbol{\sigma}_{i_1} \cdot \left( \boldsymbol{\sigma}_{i_2} \times \left( \boldsymbol{\sigma}_{i_3} \times \left( \cdots \times \left( \boldsymbol{\sigma}_{i_{n-1}} \times \boldsymbol{\sigma}_{i_n} \right) \cdots \right) \right) \right) \\ &= \boldsymbol{\sigma}_{i_1} M_{i_2} M_{i_3} \cdots M_{i_{n-1}} \boldsymbol{\sigma}_{i_n}^\top, \end{aligned} \tag{14}$$

where in the second line, $\boldsymbol{\sigma}_i = (X_i, Y_i, Z_i)$ is treated as a row vector, and $\boldsymbol{\sigma}_i^\top$ is its transpose.

For the more general SU($N$) invariant chain, the building block for the MPO becomes the $N^2 - 1$ by $N^2 - 1$ matrix defined by:

$$\left( M_i^{(N)} \right)_{ac} := \sum_{b=1}^{N^2 - 1} f^{abc} t_i^b, \tag{15}$$

where the indices run $a, c = 1, 2, \ldots, N^2 - 1$. We note that $M_i^{(N=2)} = M_i$. The nested product (8) for the SU($N$) case is represented by:

$$f_n(i_1, i_2, \ldots, i_n) = \mathbf{t}_{i_1} M_{i_2} M_{i_3} \cdots M_{i_{n-1}} \mathbf{t}_{i_n}^\top, \tag{16}$$

where $\mathbf{t}_i = (t_i^1, \ldots, t_i^{N^2 - 1})$ is treated as a row vector, and $\mathbf{t}_i^\top$ is its transpose.

We introduce another building block for the MPO, the 3 by 3 diagonal matrix $e$ for the SU(2) case and the $N^2 - 1$ by $N^2 - 1$ diagonal matrix $e^{(N)}$ for the general SU($N$) cases, the diagonal elements of which are the identity operators:

$$e := \begin{pmatrix} I & O & O \\ O & I & O \\ O & O & I \end{pmatrix}, \qquad e^{(N)} := \begin{pmatrix} I & & & \\ & I & & N^2 - 1 \\ & & \ddots & \\ & & & I \end{pmatrix}, \tag{17}$$

where the non-diagonal elements are all $O$. We note that $e^{(2)} = e$. We give the expressions for the local charges in the $L = 4$ case using the building blocks in appendix D.

### 3.3 Explicit expressions of MPO

Next, we construct the MPO for the local conserved quantities from the building block defined above. While we focus on the spin-1/2 Heisenberg chain (SU(2) case) here, the scenario for the more general SU($N$) case is similar as well: simply replace $M_i, e, \boldsymbol{\sigma}_i$ with $M_i^{(N)}, e^{(N)}, \mathbf{t}_i$, respectively.

MPO components for $k$-th local conserved quantity $Q_k$ for $k > 2$ are the upper triangle square matrices with bond dimension $3k - 1$:

$$\Gamma_k^i := \begin{pmatrix} I & \boldsymbol{\sigma}_L & & \\ & & & \\ & & \mathcal{M}_k^i + \Theta_k & \\ & & & \\ & & & \boldsymbol{\sigma}_1^\top \\ & & & I \end{pmatrix}, \tag{18}$$

where $\mathcal{M}_k^i$ and $\Theta_k$ are the square matrices of size $3(k - 2)$, and the blank blocks are entirely filled with zero-operators. $\mathcal{M}_k^i$ is defined by:

$$\mathcal{M}_k^i := \begin{pmatrix} M_i & & & k-2 \\ & M_i & & \\ & & \ddots & \\ & & & M_i \end{pmatrix}, \tag{19}$$

where the diagonal 3 by 3 block elements are all $M_i$ and the non-diagonal elements are all $O_{3,3}$. $\Theta_k$ is defined by:

$$(\Theta_k)_{a,b} := \begin{cases} C_n e & (\exists n \in \mathbb{N} : b - a = 2n + 1), \\ O_{3,3} & (\text{otherwise}), \end{cases} \tag{20}$$

where $a$ and $b$ ($1 \le a, b \le k - 2$) indicate the coordinate for the 3 by 3 block matrices in $\Theta_k$, and $\mathbb{N} = \{0, 1, 2, \ldots\}$ is the set of all natural numbers and $C_n = \binom{2n}{n} - \binom{2n}{n-1}$ is $n$-th Catalan number, which is the special case of the generalized Catalan number $C_{n,n} = C_n$. $\Theta_k$ can also be obtained from a simple recursion equation, which will be shown in the appendix C.

One can calculate the $k$-th local conserved quantity $Q_k$ by taking the product of $\Gamma_k^i$ over all the sites and doing some boundary treatment. To be specific, for $L \ge k$ one can write the explicit form of the matrix product as:

$$\Gamma_k^1 \Gamma_k^2 \cdots \Gamma_k^L = \begin{pmatrix} I & \mathbf{u}_{k,L} & Q_k^c \\ & & \\ & O_{m,m} & \mathbf{v}_{k,1}^\top \\ & & I \end{pmatrix}, \tag{21}$$

where $m \equiv 3k - 2$ and $\mathbf{u}_{k,L} = (u_{k,L}^1, \ldots, u_{k,L}^{3(k-1)})$ and $\mathbf{v}_{k,1} = (v_{k,1}^1, \ldots, v_{k,1}^{3(k-1)})$ is the row vector of $3(k - 1)$ dimension with its elements being the operator localized at the boundary. $u_{k,L}^j$ has non-trivial action on the sites from the $(L - k + 1)$-th site to the $L$-th site, and $v_{k,1}^j$ has non-trivial action from the 1-st site to the $k$-th site. $Q_k^c$ is the bulk term of the $Q_k$, which can be more simply written with boundary vector:

$$\langle L | \Gamma_k^1 \Gamma_k^2 \cdots \Gamma_k^L | R \rangle = Q_k^c, \tag{22}$$

where $\langle L| = (1, 0, \ldots, 0)$ and $|R\rangle = (0, \ldots, 0, 1)^\top$ are the boundary vectors of the same size with $\Gamma_k^i$. The local conserved quantity $Q_k$ under the periodic boundary condition is given with the boundary term $q_k^B \equiv \mathbf{u}_{k,L} \mathbf{v}_{k,1}^\top$:

$$Q_k = Q_k^c + q_k^B. \tag{23}$$

The boundary term $q_k^B$ is constructed from the operators that jump over the boundary, i.e. the linear combination of the operators that act across the boundary from the $L$-th site to the 1-st site. In the periodic boundary case considered here, the structure of the boundary terms is the same as that of the bulk term.

For the case that the Hamiltonian is defined on the infinite chain, i.e. the summations over sites $\sum_{i=1}^L$ are all replaced with $\sum_{i \in \mathbb{Z}}$ where $\mathbb{Z}$ is the set of all integers, the boundary term $q_k^B$ becomes irrelevant and it is enough to consider only the bulk term. The MPO representation for the local charge $Q_k^\infty$ in the infinite chain is simply given by:

$$Q_k^\infty = \langle L| \overrightarrow{\prod_{i \in \mathbb{Z}}} \Gamma_k^i |R\rangle, \tag{24}$$

where for the ordering of the operators in the product, we choose a convention that operators with lower site indices act first.

When taking products of $\Gamma_k^i$, we can treat each building block as if it was a non-commutative scalar because the building blocks appearing in the MPO are conformable for multiplication. Therefore, in the following, we denote the zero-operator block matrix $O_{n,m}$ just simply as $O$.

We give the expressions up to $\Gamma_8^i$ below:

$$\Gamma_2^i = \begin{pmatrix} I & \boldsymbol{\sigma}_i & O \\ O & \boldsymbol{\sigma}_i^\top \\ & I \end{pmatrix},$$

$$\Gamma_3^i = \begin{pmatrix} I & \boldsymbol{\sigma}_i & O & O \\ O & M_i & O \\ & O & \boldsymbol{\sigma}_i^\top \\ & & I \end{pmatrix},$$

$$\Gamma_4^i = \begin{pmatrix} I & \boldsymbol{\sigma}_i & O & O & O \\ O & M_i & C_0 e & O \\ & O & M_i & O \\ & & O & \boldsymbol{\sigma}_i^\top \\ & & & I \end{pmatrix},$$

$$\Gamma_5^i = \begin{pmatrix} I & \boldsymbol{\sigma}_i & O & O & O & O \\ O & M_i & C_0 e & O & O \\ & O & M_i & C_0 e & O \\ & & O & M_i & O \\ & & & O & \boldsymbol{\sigma}_i^\top \\ & & & & I \end{pmatrix},$$

$$\Gamma_6^i = \begin{pmatrix} I & \boldsymbol{\sigma}_i & O & O & O & O & O \\ O & M_i & C_0 e & O & C_1 e & O \\ & O & M_i & C_0 e & O & O \\ & & O & M_i & C_0 e & O \\ & & & O & M_i & O \\ & & & & O & \boldsymbol{\sigma}_i^\top \\ & & & & & I \end{pmatrix}, \quad \Gamma_7^i = \begin{pmatrix} I & \boldsymbol{\sigma}_i & O & O & O & O & O & O \\ O & M_i & C_0 e & O & C_1 e & O & O \\ & O & M_i & C_0 e & O & C_1 e & O \\ & & O & M_i & C_0 e & O & O \\ & & & O & M_i & C_0 e & O \\ & & & & O & M_i & O \\ & & & & & O & \boldsymbol{\sigma}_i^\top \\ & & & & & & I \end{pmatrix},$$

$$\Gamma_8^i = \begin{pmatrix} I & \boldsymbol{\sigma}_i & O & O & O & O & O & O & O \\ O & M_i & C_0 e & O & C_1 e & O & C_2 e & O \\ & O & M_i & C_0 e & O & C_1 e & O & O \\ & & O & M_i & C_0 e & O & C_1 e & O \\ & & & O & M_i & C_0 e & O & O \\ & & & & O & M_i & C_0 e & O \\ & & & & & O & M_i & O \\ & & & & & & O & \boldsymbol{\sigma}_i^\top \\ & & & & & & & I \end{pmatrix},$$

where the lower triangular elements are all zero-operators and the $O$'s in these expressions represent zero-operator block matrices, suitably shaped for the position of their respective blocks.

In the following section, we treat the building block elements of $\Gamma_k^i$ such as $M_i$ and $\boldsymbol{\sigma}$ and $e$ and $O$ as a symbol, and the indices indicate the positions of these building block elements. For example, $\left(\Gamma_k^i\right)_{1,2} = \boldsymbol{\sigma}_i$, and $\left(\Gamma_k^i\right)_{2,3} = M_i$ for $k \geq 3$, and $\left(\Gamma_8^i\right)_{2,8} = C_2 e$.

We note that $\Gamma_k^i$ is only involved with the usual Catalan number, while the expressions for the local charges themselves (4) are also involved with the generalized Catalan number. In this point, the MPO representation (18) is more simplified than the traditional bare expressions (4).

We briefly explain why the local charges can be represented simply by the MPO. The point is that basic components of the local charges (2) or (8) are the nested product of the Pauli matrices, and consequently, we can rewrite them by the products of $M_i$ like (14) or (16). We can then factorize them into the form of an MPO, as shown in (21). However, this does not apply straightforwardly to the anisotropic generalizations: basic components of the local charges for the XXZ or XYZ chains do not have this nested product structure [9,10], and they cannot be easily expressed in a matrix product form similar to (14) and (16). We leave the anisotropic generalization of our result to the future problem.

### 3.4 Boost operator

One may think we can obtain the recursive relation between $\Gamma_k^i$ and $\Gamma_{k+1}^i$ using the Boost operator [30]. In this subsection, we investigate the recursive way to obtain $\Gamma_k^i$.

For simplicity, we consider the infinite chain here. Within this subsection, we denote the local charges in the infinite chain $Q_k^\infty$ simply by $Q_k$.

The boost operator in the infinite chain is defined by:

$$B := \sum_{j \in \mathbb{Z}} j \boldsymbol{\sigma}_j \cdot \boldsymbol{\sigma}_{j+1}, \tag{25}$$

which generate the recursive relation between $Q_k$ and $Q_{k+1}$:

$$[Q_k, B] = Q_{k+1}. \tag{26}$$

Strictly speaking, the local charge obtained from the boost operation is different from our $Q_k$ defined in (4); local charges have the freedom to add the lower-order charges, and the boost-derived charge and our $Q_k$ exhibit distinct linear combinations of lower-order charges. Thus, $Q_k$ in (26) is slightly different from its original definition (4), and the corresponding MPO representation is also slightly different from (18). However, the bond dimension of the MPO for boost-derived charges seems to be still the same as that of (18). Within this subsection, we denote boost-derived charges by $Q_k$ and its MPO by $\Gamma_k^j$.

The MPO representation for the boost operator is given by:

$$K^j = \begin{pmatrix} I & j\boldsymbol{\sigma}_j & O \\ & & \boldsymbol{\sigma}_j^\top \\ & O_{4,4} & \\ & & I \end{pmatrix}, \tag{27}$$

where the bond dimension is 5, which is the same as that of the Hamiltonian, and the boost operator is written as:

$$B = \langle L_2 | \overrightarrow{\prod_{i \in \mathbb{Z}}} K^i | R_2 \rangle, \tag{28}$$

where we denote $\langle L_n| = (1, 0, \ldots, 0)$ and $|R_n\rangle = (0, \ldots, 0, 1)^\top$ with $0, \ldots, 0$ representing the sequence of $3n - 2$ zeros.

We denote the local physical space corresponding to $j$-th site by $h_j \cong \mathbb{C}^2$, and denote the auxiualy space of $\Gamma_k^j$, corresponding to the bond dimension $3k - 1$, by $V_{a_1} \cong \mathbb{C}^{3k-1}$. Similary, we denote the auxiualy space of $K^j$, corresponding to the bond dimension 5, by $V_{a_2} \cong \mathbb{C}^5$. We define the Hilbert space $\mathcal{W} \equiv \left(\bigotimes_{j\in\mathbb{Z}} h_j\right) \otimes V_{a_1} \otimes V_{a_2}$.

We can treat $\Gamma_k^j$ as an operator on $\mathcal{W}$, acting non-trivially on $h_j$ and $V_{a_1}$, and $K^j$ as an operator on $\mathcal{W}$, acting non-trivially on $h_j$ and $V_{a_2}$. Given this context, we denote $\Gamma_k^j$ and $K^j$ as $\Gamma_k^{j,a_1}$ and $K^{j,a_2}$, respectively.

With these notations and (24), we have:

$$Q_k B = {}_a\langle L| \left( \overrightarrow{\prod_{j\in\mathbb{Z}}} \Gamma_k^{j,a_1} K^{j,a_2} \right) |R\rangle_a, \qquad BQ_k = {}_a\langle L| \left( \overrightarrow{\prod_{j\in\mathbb{Z}}} K^{j,a_2} \Gamma_k^{j,a_1} \right) |R\rangle_a, \qquad (29)$$

where ${}_a\langle L| \equiv {}_{a_1}\langle L_k| \otimes {}_{a_2}\langle L_2|$ and $|R\rangle_a \equiv |R_2\rangle_{a_2} \otimes |R_k\rangle_{a_1}$ are vectors of dimension $5(3k-1)$, and a vector with the indices $a_{1(2)}$ denotes a vector in $V_{a_{1(2)}}$. We can represent the boost recursion relation (26) as:

$$\langle \overline{L}| \overrightarrow{\prod_{j\in\mathbb{Z}}} \overline{\Gamma}_{k+1}^j |\overline{R}\rangle = Q_{k+1}, \qquad (30)$$

where we denote $\langle \overline{L}| \equiv ( {}_a\langle L|, {}_a\langle L| )$, and $|\overline{R}\rangle \equiv ( |R\rangle_a, -|R\rangle_a )^\top$, which are vectors of dimension $10(3k-1)$, and

$$\overline{\Gamma}_{k+1}^j \equiv \begin{pmatrix} \Gamma_k^{j,a_1} K^{j,a_2} & O \\ O & K^{j,a_2} \Gamma_k^{j,a_1} \end{pmatrix}, \qquad (31)$$

where the elements of two-by-two matrix $\overline{\Gamma}_{k+1}^j$ is the operator on $\mathcal{W}$, and $O$ is zero-operator on $\mathcal{W}$, and the bond dimension of $\overline{\Gamma}_{k+1}^j$ is $10(3k-1)$.

In this way, we can derive the MPO representation for $Q_{k+1}$ as $\overline{\Gamma}_{k+1}^j$ from the boost recursion relation (26). However, $\overline{\Gamma}_{k+1}^j$ is different from $\Gamma_{k+1}^j$: the bond dimension of $\overline{\Gamma}_{k+1}^j$ is $10(3k-1)$ whereas $\Gamma_{k+1}^j$ possesses a bond dimension of $3(k+1)-1$, which is smaller than that of $\overline{\Gamma}_{k+1}^j$. Thus, $\Gamma_{k+1}^j$ is simpler than $\overline{\Gamma}_{k+1}^j$. Though it might be feasible to derive the expression for $\Gamma_{k+1}^j$ from $\overline{\Gamma}_{k+1}^j$ through the compression of the MPO, addressing this is beyond our current study.

# 4 Proof of the MPO

In this section, we give the demonstration that the MPO introduced in the previous section actually reproduces the local conserved quantities of the spin-1/2 Heisenberg chain.

## 4.1 Catalan number identity

There is the useful identity of generalized Catalan numbers $C_{n+m,n}$ and usual Catalan numbers $C_n$ for the proof of the MPO:

$$C_{n+m,n} = \sum_{\substack{n_0 + \cdots + n_m = n \\ n_j \geq 0}} C_{n_0} \cdots C_{n_m} = \sum_{\substack{\sum_{j=0}^m n_j = n \\ n_j \geq 0}} \prod_{j=0}^m C_{n_j}. \qquad (32)$$

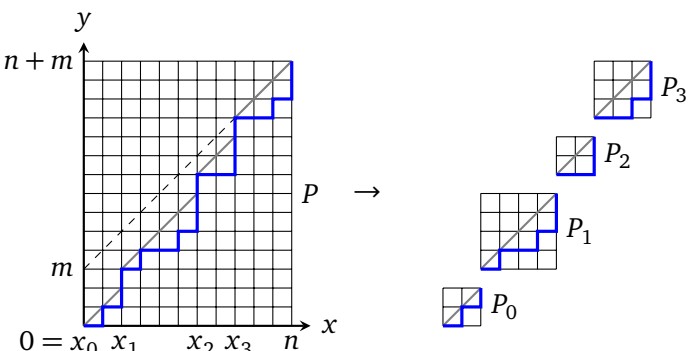

Figure 1: Graphical explanation of the proof of the recurrence relations of generalized Catalan numbers eq. (32). $n = 11$, $m = 3$ case is shown. $C_{n+m,n}$ equals the number of possible lattice paths from $(0,0)$ to $(n+m,n)$ that never crosses the line $y = x+m$. If one divides a path $P$ into $(m+1)$ pieces $P_0, P_1, \ldots, P_m$ by the point where $P$ first reaches the line $y = x + j$, each $P_j$ can be related one by one to a possible lattice path from $(0,0)$ to $(x_{j+1} - x_j, x_{j+1} - x_j)$ that never reaches the line $y = x$.

One can prove this recurrence relation (32) by relating $C_{n+m,n}$ to the numbers of monotone lattice paths from $(x, y) = (0,0)$ to $(n, n+m)$ that never crosses the line $y = x + m$.

Let $\Omega_{n+m,n}$ represent the set of monotone lattice paths that meet this condition. For each $P \in \Omega_{n+m,n}$, let $x_j(P)$ be the smallest $x$ coordinate of the point on the path where the path reaches the line $y = x + j$ for the first time for $1 \leq j \leq m$. We define $x_0(P) \equiv 0$ and $x_{m+1}(P) \equiv n$. We define the subset of $\Omega_{n+m,n}$ as follows:

$$\Omega_{n+m,n}^{n_0,\ldots,n_m} := \left\{ P \in \Omega_{n+m,n} \middle| \forall j \in \mathbb{N}, 0 \leq j \leq m : x_{j+1}(P) - x_j(P) = n_j \right\}. \tag{33}$$

$\Omega_{n+m,n}$ can be represented by the direct sum of $\Omega_{n+m,n}^{n_0,\ldots,n_m}$:

$$\Omega_{n+m,n} = \bigsqcup_{\substack{n_0+\cdots+n_m=n \\ n_j \geq 0}} \Omega_{n+m,n}^{n_0,\ldots,n_m}. \tag{34}$$

Let us denote $P_j$ as the part of a path $P$ from $(x_j, x_j)$ to $(x_{j+1}, x_{j+1})$, corresponding to the element of $\Omega_{n_j,n_j}$, that is, to a monotone lattice path from $(0,0)$ to $(n_j, n_j)$ that never crosses the line $y = x$ (see Fig. 1). Because the number of elements in $\Omega_{n_j,n_j}$ equals $C_{n_j}$, one gets

$$\#\left[ \Omega_{n+m,n}^{n_0,\ldots,n_m} \right] = \prod_{j=0}^{m} C_{n_j}, \tag{35}$$

where $\#[\bullet]$ denotes the number of elements in a (finite) set. Using (34), we have

$$C_{n+m,n} = \#\left[ \Omega_{n+m,n} \right] = \sum_{\substack{\sum_{j=0}^{m} n_j = n \\ n_j \geq 0}} \#\left[ \Omega_{n+m,n}^{n_0,\ldots,n_m} \right] = \sum_{\substack{\sum_{j=0}^{m} n_j = n \\ n_j \geq 0}} \prod_{j=0}^{m} C_{n_j}. \tag{36}$$

Thus, we have proved (32).

After the completion of this work, we became aware of the conference proceedings [31], where the identity (32) is discussed. While they derive the identity analytically, we derive the identity combinatorially with the lattice path explanation. A formula similar to but distinct from ours has also been studied in [32], where the summation of indices traverses $n_j \geq 1$.

## 4.2  Demonstration of equivalence of MPO and $Q_k$

In this subsection, we demonstrate the MPO actually reproduces the local conserved quantities, taking the case of $Q_{12}$ as an example. The rigorous proof is given in appendix A for the bulk term and in appendix B for the boundary term.

Considering Eq. (4), we can see the operator $\boldsymbol{\sigma}_1 M_2 M_4 \boldsymbol{\sigma}_6^\top$ is included in $F_{12-2(n+m),m} = F_{4,2}$ for $(n, m) = (2, 2)$, and therefore included in $Q_{12}$, particularly being included in the bulk term $Q_{12}^c$ for $L > 6$. The coefficient of $\boldsymbol{\sigma}_1 M_2 M_4 \boldsymbol{\sigma}_6^\top$ in $Q_{12}$ is $C_{n+m-1,n} = C_{3,2}$. In the following, we demonstrate this coefficient and operator are reproduced by the MPO made from $\Gamma_{12}^i$.

$Q_{12}^c$ is calculated as follows:

$$Q_{12}^c = \left( \Gamma_{12}^1 \Gamma_{12}^2 \cdots \Gamma_{12}^L \right)_{1,13} = \sum_{\substack{1 = a_1 \le a_2 \le \cdots \\ \cdots \le a_L \le a_{L+1} = 13}} \gamma_{a_1, a_2}^1 \gamma_{a_2, a_3}^2 \cdots \gamma_{a_L, a_{L+1}}^L \,, \tag{37}$$

where the indices indicate the position of the building blocks, and $\gamma_{a_p, a_{p+1}}^p$ represents the $(a_p, a_{p+1})$-th block element of $\Gamma_{12}^p$, and the second equality holds from the fact that $\Gamma_{12}^p$ is an upper triangular matrix. In the summation in (37), $\boldsymbol{\sigma}_1 M_2 M_4 \boldsymbol{\sigma}_6^\top$ is included in the restricted summation of (37) as follows:

$$\sum_{1 = a_1 \le a_2 \le \cdots \le a_6 \le a_7 = 13} \gamma_{a_1, a_2}^1 \gamma_{a_2, a_3}^2 \gamma_{a_3, a_4}^3 \gamma_{a_4, a_5}^4 \gamma_{a_5, a_6}^5 \gamma_{a_6, a_7}^6 \,, \tag{38}$$

where the other variables in (37) is fixed as $a_p = 13$ for $p \ge 7$, and therefore $\gamma_{a_p, a_{p+1}}^p = I$ for $p \ge 7$.

Each term in (38) corresponds to the "path" from "start" to "end" traversing through the matrix in Figure 2, which has the same structure as $\Gamma_{12}^i$. A "path" is defined as follows: first, we pick up the element in the first row, where only $\boldsymbol{\sigma}$ is non-zero.[2] Thus we have to pick $\boldsymbol{\sigma}$ first, corresponding to $\gamma_{1,2}^1 = \boldsymbol{\sigma}_1$. Every time we pick an element, we then vertically descend within the same column till we reach the diagonal line, after which we move horizontally to the right within the same row and pick up an element on the row. The element chosen in our $p$-th pick corresponds to $\gamma_{a_p, a_{p+1}}^p$ in equation (38). At the final (sixth) pick, we have to pick $\boldsymbol{\sigma}^\top$, corresponding to $\gamma_{a_6, a_7}^6 = \boldsymbol{\sigma}_6^\top$, and finish the procedure for this path. By considering all possible paths, we can compute equation (38). For calculating the coefficient of $\boldsymbol{\sigma}_1 M_2 M_4 \boldsymbol{\sigma}_6^\top$ in $Q_{12}^c$, we have to pick up the point of $\boldsymbol{\sigma}, M, C_{n_1}e, M, C_{n_2}e, \boldsymbol{\sigma}^\top$ in the matrix in Figure 2 in that order. There are three possible paths that generate $\boldsymbol{\sigma}_1 M_2 M_4 \boldsymbol{\sigma}_6^\top$ in $Q_{12}^c$, which are depicted by the teal, the bold blue, and the purple line in Figure 2, corresponding to $(n_1, n_2) = (0, 2)$, $(1, 1)$, and $(2, 0)$ respectively. These paths all satisfy $n_1 + n_2 = 2$. All the contribution to the coefficient is

$$C_0 C_2 + C_1 C_1 + C_2 C_0 = C_{3,2} \,, \tag{39}$$

where we used the Catalan number identity (32). Thus, we have demonstrated that the coefficients of $\boldsymbol{\sigma}_1 M_2 M_4 \boldsymbol{\sigma}_6^\top$ in the MPO is actually $C_{3,2}$, which is the same as the case of the local conserved quantity $Q_{12}$.

Let us consider a more general situation, leaving the rigorous proof to the appendix A. We calculate the coefficient of the operator $\boldsymbol{\sigma}_{i_1} M_{i_2} \cdots M_{i_{j-1}} \boldsymbol{\sigma}_{i_j}^\top$ in $Q_k^c$ for $j = k - 2(n + m)$ and with $m$-holes, i.e. $i_j - i_1 + 1 - j = m$. The paths that generate $\boldsymbol{\sigma}_{i_1} M_{i_2} \cdots M_{i_{j-1}} \boldsymbol{\sigma}_{i_j}^\top$ have to pick up the elements of $C_{n_1}e, \ldots, C_{n_m}e$, corresponding to the $m$-holes. Every time we pick up $C_n e$ on the paths, there is a horizontal move of $(2n + 2)$ columns to the right, and every

---

[2]Indeed, $(1, 1)$-component of $\Gamma_k^i$ is $I$, presenting another non-zero component on the first row. Nevertheless, an initial pick of $I$ does not yield $\boldsymbol{\sigma}_1 M_2 M_4 \boldsymbol{\sigma}_6^\top$ in $Q_{12}^c$, thus these cases are not considered here.

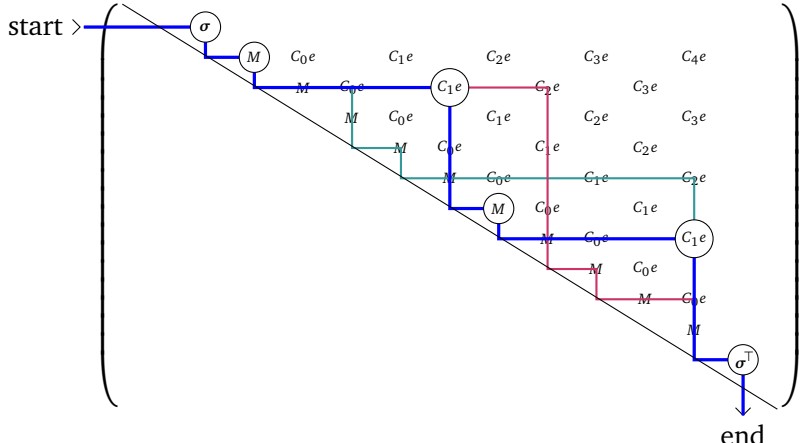

Figure 2: An intuitive method for calculating the coefficient of $\sigma_1 M_2 M_4 \sigma_6^\top$ in $Q_{12}^c$, which is one of components of $F_{4,2}$. The three paths depicted by the teal, the bold blue, and the purple line contribute to the coefficients. For example, the bold blue line represents the path that picks $\sigma$, $M$, $C_1 e$, $M$, $C_1 e$, $\sigma^\top$ in this order, and the contribution is $C_1{}^2 \sigma_1 M_2 M_4 \sigma_6^\top$. To obtain the coefficients of $F_{k-2(n+m),m}$ in more general $Q_k^c$, we must take all the possible paths including $C_{n_1} e, \ldots, C_{n_m} e$ in this order that satisfy $n_1 + \cdots + n_m = n$.

time we pick up $M$, there is a horizontal move of one column to the right. The number of the columns between $\sigma$ and $\sigma^\top$ of the matrix corresponding to $\Gamma_k^i$ is $k-2$, thus we have the relation $\sum_{p=1}^m (2n_p + 2) + j - 2 = k - 2$. Solving this equation, we have $n_1 + \cdots + n_m = n$, and all the contributions to the coefficients become

$$\sum_{\substack{n_1+n_2+\cdots+n_m=n \\ n_j \geq 0}} C_{n_1} C_{n_2} \cdots C_{n_m} = C_{n+m-1,n}, \tag{40}$$

where we used the Catalan number identity (32), and we can see the coefficient of (4) is reproduced by the MPO.

## 5 Summary and Outlook

In this work, we show the matrix product operator(MPO) representation of the local conserved quantities for the spin-1/2 Heisenberg chain and its SU($N$) generalization. In terms of our MPO representation, the local conserved quantities are more simply written: the pattern in the expression of the MPO is more straightforward than that of the local conserved quantities themselves. Especially the coefficients appearing in the MPO are simpler than those in the bare expressions for the local charges (4), and the Catalan tree pattern of the local conserved quantities naturally appears from the product of the MPOs. This means the complexity of the local conserved quantities is folded into the product of the MPO.

The generalization of our result to the local charges in the spin-1/2 XYZ chain [9], which is the anisotropic generalization of the Heisenberg chain considered here is an interesting topic. It would also be worthwhile to verify how the mutual commutativity of the local conserved quantities can be explained with our MPO representation.

Our strategy may be useful to find the general expressions for the local charges in more general integrable quantum spin chains. To obtain the general expression for the local charges, one must identify the operator basis that constructs them and discern the regularity of their

coefficients. When looking at the bare local charges, these tasks are very hard for interacting integrable spin chains. Hence, once one represents the lower-order charges with MPO, the patterns may become clearer and we may predict the general forms for the local charges via MPO representation. The local charges considered in this work have already been found for decades before. Therefore, it is interesting to investigate the MPO representation for local charges whose general forms have not been found. In this spirit, the investigation of local charges in open boundary cases is also interesting. In the periodic boundary condition, the boundary term of the local conserved quantity is trivial: they have the same structure as the bulk term, as discussed in Appendix B. However, the boundary term in the open boundary condition is non-trivial, and the expressions have yet to be elucidated even for the most famous spin-1/2 Heisenberg chain [33]. Using the MPO representation, we might infer the pattern of the boundary terms in the open boundary case.

## Acknowledgments

We thank Hosho Katsura for his helpful comment at the JPS 2023 Spring Meeting and his insightful comment on our manuscript. We thank Yoshitaka Okuyama for the fruitful discussion.

**Funding information** K. F. was supported by Forefront Physics and Mathematics Program to Drive Transformation (FoPM), a World-leading Innovative Graduate Study (WINGS) Program, and JSR Fellowship, the University of Tokyo, and KAKENHI Grants No. JP21J20321 from the Japan Society for the Promotion of Science (JSPS).

## A Rigorous proof for bulk term

In appendix A, we give the rigorous proof of the bulk part of (23).

In the following, the matrix indices indicate the position of the building blocks, i.e., treating the elements of $\Gamma_k^i$ such as $M_i$ and $\boldsymbol{\sigma}$ and $e$ and $O$ in $\Gamma_k^i$ as a symbol.

Calculating the product of the MPO explicitly, we have:

$$Q_k^c = \left( \Gamma_k^1 \Gamma_k^2 \cdots \Gamma_k^L \right)_{1,k+1} = \sum_{1=a_1 \leq a_2 \leq \cdots \leq a_{L-1} \leq a_L = k+1} \overrightarrow{\prod_{1 \leq p \leq L}} \gamma_{a_p,a_{p+1}}^p , \tag{A.1}$$

where the second equality holds from the fact that $\Gamma_k^i$ is an upper triangular matrix, and we denote the matrix element of $\Gamma_k^i$ by $\left( \Gamma_k^i \right)_{a,b} \equiv \gamma_{a,b}^i$. Considering the first row and the last column of $\Gamma_k^p$ is written by:

$$\gamma_{1,a}^p = \begin{cases} I & (a=1), \\ \boldsymbol{\sigma}_p & (a=2), \\ O & (\text{otherwise}), \end{cases} \qquad \gamma_{a,k+1}^p = \begin{cases} I & (a=k+1), \\ \boldsymbol{\sigma}_p^\top & (a=k), \\ O & (\text{otherwise}), \end{cases} \tag{A.2}$$

and $\gamma^p_{a,a} = O$ for $2 \leq a \leq k$, the summation in (A.1) can be decomposed as:

$$
\begin{aligned}
\text{(A.1)} &= \sum_{1 \leq i < j \leq L} \sum_{2 = a_{i+1} < a_{i+2} < \cdots < a_{j-1} < a_j = k} \boldsymbol{\sigma}_i \left( \overrightarrow{\prod_{i+1 \leq p \leq j-1}} \gamma^p_{a_p, a_{p+1}} \right) \boldsymbol{\sigma}_j^\top \\
&= \sum_{1 \leq i < j \leq L} \sum_{\substack{b_{i+1} + \cdots + b_{j-1} = k-2 \\ b_p > 0}} \boldsymbol{\sigma}_i \left( \overrightarrow{\prod_{i+1 \leq p \leq j-1}} \gamma^p_{b_p} \right) \boldsymbol{\sigma}_j^\top \\
&= \sum_{1 \leq i \leq L-k+1} \boldsymbol{\sigma}_i \gamma_1^{i+1} \gamma_1^{i+2} \cdots \gamma_1^{i+k-2} \boldsymbol{\sigma}_{i+k-1}^\top + \sum_{1 \leq i < j \leq L} \sum_{\substack{b_{i+1} + \cdots + b_{j-1} = k-2 \\ b_p > 0, \, \exists p' : b_{p'} > 1}} \boldsymbol{\sigma}_i \left( \overrightarrow{\prod_{i+1 \leq p \leq j-1}} \gamma^p_{b_p} \right) \boldsymbol{\sigma}_j^\top ,
\end{aligned}
$$

(A.3)

where the second summation on the first row is taken over $a_{i+2}, \ldots, a_{j-1}$ and the other $a_p$'s are fixed by $a_1, \ldots, a_i = 1, a_{i+1} = 2, a_j = k, a_{j+1}, \ldots a_L = k+1$, and in the second equality we define $b_p \equiv a_{p+1} - a_p$ for $i < p < j$ and $\gamma^p_{b_p} \equiv \gamma^p_{a_p, a_{p+1}}$. $\gamma^p_{b_p}$ is well-defined because $\gamma^p_{a_p, a_{p+1}}$ depends only on the difference $a_{p+1} - a_p$. In the last equality, we decompose the summation to the case that all $b_p = 1$ for all $p$ and the other cases that at least one $b_p$ is greater than one. The explicit form of $\gamma^p_{b_p}$ is:

$$
\gamma^p_{b_p} = \begin{cases} M_p & (b_p = 1), \\ C_{b_p/2-1} e & (b_p \text{ is even}), \\ O & (\text{otherwise}). \end{cases}
$$

(A.4)

The first term of (A.3) becomes

$$
\begin{aligned}
\text{First term of (A.3)} &= \sum_{1 \leq i \leq L-k+1} \boldsymbol{\sigma}_i \gamma_1^{i+1} \gamma_1^{i+2} \cdots \gamma_1^{i+k-2} \boldsymbol{\sigma}_{i+k-1}^\top \\
&= \sum_{1 \leq i \leq L-k+1} \boldsymbol{\sigma}_i M_{i+1} M_{i+2} \cdots M_{i+k-2} \boldsymbol{\sigma}_{i+k-1}^\top \\
&= \sum_{\mathcal{C} \in \mathcal{C}^{k,0}_{\text{bulk}}} f_k(\mathcal{C}) = F^{\text{bulk}}_{k,0} ,
\end{aligned}
$$

(A.5)

where $\mathcal{C}^{n+m,m}_{\text{bulk}}$ denotes the set of clusters $\mathcal{C} = \{i_1, \ldots, i_n\}$ of order $n$ with $m$ holes, satisfying $1 \leq i_1 < i_n \leq L$, and we can see $F^{\text{bulk}}_{k,0}$ is the bulk term of $F_{k,0}$.

We denote the number of $p$'s that satisfies $b_p > 1$ in the variable $\{b_{i+1}, \ldots, b_{j-1}\}$ of the second summation of (A.3) by $m$. Then the second term of (A.3) becomes:

Second term of (A.3)

$$
\begin{aligned}
&= \sum_{1 \leq i < j \leq L} \sum_{\substack{b_{i+1} + \cdots + b_{j-1} = k-2 \\ b_p > 0, \, \exists p' : b_{p'} > 1}} \boldsymbol{\sigma}_i \left( \overrightarrow{\prod_{i < p < j}} \gamma^p_{b_p} \right) \boldsymbol{\sigma}_j^\top \\
&= \sum_{1 \leq i < j \leq L} \sum_{m=1}^{w-2} \sum_{i < i_1 < \cdots < i_{w-2-m} < j} \sum_{\substack{b_{i'_1} + \cdots + b_{i'_m} = k-2-(w-2-m) \\ b_{i'_{m'}} > 1}} \boldsymbol{\sigma}_i \left( \prod_{r=1}^m \gamma^{i'_r}_{b_{i'_r}} \right) \left( \overrightarrow{\prod_{1 \leq r \leq w-2-m}} \gamma^{i_r}_{b_{i_r}} \right) \boldsymbol{\sigma}_j^\top \\
&= \sum_{1 \leq i < j \leq L} \sum_{m=1}^{w-2} \left( \sum_{\substack{b_{i'_1} + \cdots + b_{i'_m} = k-w+m \\ b_{i'_{m'}} > 1}} \prod_{r=1}^m \gamma_{b_{i'_r}} \right) \sum_{i < i_1 < \cdots < i_{w-2-m} < j} \boldsymbol{\sigma}_i M_{i_1} M_{i_2} \cdots M_{i_{w-2-m}} \boldsymbol{\sigma}_j^\top
\end{aligned}
$$

$$
\begin{aligned}
&= \sum_{w=2}^{k} \sum_{m=1}^{w-2} \left( \sum_{\substack{c_1+\cdots+c_m=k-w+m \\ c_r>1}} \prod_{r=1}^{m} \gamma_{c_r} \right) \sum_{i=1}^{L-w-1} \sum_{i<i_1<\cdots<i_{w-2-m}<i+w-1} \boldsymbol{\sigma}_i M_{i_1} M_{i_2} \cdots M_{i_{w-2-m}} \boldsymbol{\sigma}_{i+w-1}^{\top} \\
&= \sum_{w=2}^{k} \sum_{m=1}^{w-2} \left( \sum_{\substack{c_1+\cdots+c_m=k-w+m \\ c_r>1}} \prod_{r=1}^{m} \gamma_{c_r} \right) \sum_{\mathcal{C}\in\mathcal{C}_{\text{bulk}}^{w-m,m}} f_{w-m}(\mathcal{C}) \\
&= \sum_{w=2}^{k} \sum_{m=1}^{w-2} \left( \sum_{\substack{2(d_1+\cdots+d_m)=k-w+m \\ d_r>1}} \prod_{r=1}^{m} \gamma_{2d_r} \right) \sum_{\mathcal{C}\in\mathcal{C}_{\text{bulk}}^{w-m,m}} f_{w-m}(\mathcal{C}) \\
&= \sum_{n=0}^{\lfloor k/2\rfloor} \sum_{m=1}^{\lfloor k/2\rfloor-n} \left( \sum_{\substack{d_1+\cdots+d_m=n+m \\ d_r\geq 1}} \prod_{r=1}^{m} C_{d_r-1} \right) \sum_{\mathcal{C}\in\mathcal{C}_{\text{bulk}}^{k-2(n+m),m}} f_{k-2(n+m)}(\mathcal{C}) \\
&= \sum_{\substack{0<n+m<\lfloor k/2\rfloor \\ n\geq 0,m\geq 1}} \left( \sum_{\substack{d_1+\cdots+d_m=n \\ d_r\geq 0}} \prod_{r=1}^{m} C_{d_r} \right) F_{k-2(n+m),m}^{\text{bulk}} \\
&= \sum_{\substack{0<n+m<\lfloor k/2\rfloor \\ n\geq 0,m\geq 1}} C_{n+m-1,n} F_{k-2(n+m),m}^{\text{bulk}} ,
\end{aligned}
\tag{A.6}
$$

where we denote $w \equiv j-i+1$ and we decompose the summation $\sum_{b_{i+1}+\cdots+b_{j-1}=k-2,b_p>0}$, after fixing $m$, to the summation over $\{i_1,\ldots,i_{w-1-m}\}$ where $i_r$ satisfies $b_{i_r}=1$ and the summation over $\left\{ b_{i'_1},\ldots,b_{i'_{w_m}} \right\}$ where $i'_r$ satisfies $b_{i'_r}>1$. Note that

$$
\{i'_1,\ldots,i'_m\} \bigcup \{i_1,\ldots,i_{w-2-m}\} = \{i+1,\ldots,j-1\} .
$$

In the third equality, we used the following relation:

$$
\boldsymbol{\sigma}_i \left( \prod_{r=1}^{m} \gamma_{b_{i'_r}}^{i'_r} \right) \left( \overrightarrow{\prod_{1\leq r\leq w-2-m}} \gamma_{b_{i_r}}^{i_r} \right) \boldsymbol{\sigma}_j^{\top} = \left( \prod_{r=1}^{m} \gamma_{b_{i'_r}} \right) \boldsymbol{\sigma}_i M_{i_1} M_{i_2} \cdots M_{i_{w-2-m}} \boldsymbol{\sigma}_{i+w-1}^{\top} ,
$$

where $\gamma_{b_p}$ is defined by $\gamma_{b_p} \equiv C_{b_p/2-1}(0)$ for $b_p$ is even(odd), and the factor on the left hand side $\left( \prod_{r=1}^{m} \gamma_{b_{i'_r}}^{i'_r} \right)$ can be treat as a $c$-number, and factored out on the right hand side because $\gamma_{b_{i'_r}}^{i'_r}$ is $O$ or is proportional to $e$ for $b_{i'_r}>1$, and $e$ behave as identity operator: $eM_p = M_p e = M_p$ and $\boldsymbol{\sigma}_i e = \boldsymbol{\sigma}_i$. In the fourth equality, we replace the variable by $c_r = b_{i'_r}$. If $c_r$ is odd for some $r$, we can see $\gamma_{c_r}=0$. Thus the non-zero contribution comes from the case that all $c_r$ is even, and in the sixth equality, we rewrite $c_r$ by $c_r = 2d_r$. For $k-w+m$ being an even integer, $w$ and $m$ have to satisfy the relation $w = k-2n-m$ with $\exists n \in \mathbb{N}$. In the seventh equality, we change the variable by $d_r - 1 \to d_r$, and in the ninth equality, we used the Catalan number identity (32). $F_{k-2(n+m),m}^{\text{bulk}}$ is the bulk term of $F_{k-2(n+m),m}$.

Therefore, with (A.5) and (A.6), we have proved $Q_k^c$ becomes:

$$
Q_k^c = F_{k,0}^{\text{bulk}} + \sum_{\substack{0<n+m<\lfloor k/2\rfloor \\ n\geq 0,m\geq 1}} C_{n+m-1,n} F_{k-2(n+m),m}^{\text{bulk}} ,
\tag{A.7}
$$

and we can see this is actually the bulk term of $Q_k$.

# B  Boundary treatment

We prove the boundary terms of the local conserved quantities $Q_k$ are given by $q_k^{\mathrm{B}} \equiv \mathbf{u}_{k,L}\mathbf{v}_{k,1}^{\top}$.

We write the $k$-th matrix product operator from $i$-th site to $j$-th site ($j - i \geq k$) by:

$$\Gamma_k^i \Gamma_k^{i+1} \cdots \Gamma_k^j = \begin{pmatrix} I & \mathbf{u}_{k,j} & Q_{k,[i:j]}^{\mathrm{c}} \\ & & \mathbf{v}_{k,i}^{\top} \\ & O & \\ & & I \end{pmatrix}, \tag{B.1}$$

where $\mathbf{u}_{k,j}$ and $\mathbf{v}_{k,i}$ are the row vectors of dimension $3(k-1)$, whose elements are independent of $i$ and $j$, respectively, and are constructed from the local operators that act on at most $k$ adjacent sites between the $(j-k+1)$-th site and $j$-th site and between the $i$-th site and $(i+k-1)$-th site, respectively. $\mathbf{u}_{k,j}(\mathbf{v}_{k,i})$ is independent of $i(j)$. $Q_{k,[i:j]}^{\mathrm{c}}$ is constructed from the local operators that act on at most $k$ adjacent sites between the $i$-th site and the $j$-th site.

We can decompose the products in the MPO for the $k$-th local conserved quantities as:

$$\Gamma_k^1 \Gamma_k^2 \cdots \Gamma_k^{L+N} = \Gamma_k^1 \Gamma_k^2 \cdots \Gamma_k^L \times \Gamma_k^{L+1} \Gamma_k^{L+2} \cdots \Gamma_k^{L+N}$$

$$= \begin{pmatrix} I & \mathbf{u}_{k,L} & Q_{k,[1:L]}^{\mathrm{c}} \\ & & \mathbf{v}_{k,1}^{\top} \\ & O & \\ & & I \end{pmatrix} \begin{pmatrix} I & \mathbf{u}_{k,L+N} & Q_{k,[L+1:L+N]}^{\mathrm{c}} \\ & & \mathbf{v}_{k,L+1}^{\top} \\ & O & \\ & & I \end{pmatrix}, \tag{B.2}$$

and we have

$$Q_{k,[1:L+N]}^{\mathrm{c}} = Q_{k,[1:L]}^{\mathrm{c}} + Q_{k,[L+1:L+N]}^{\mathrm{c}} + \mathbf{u}_{k,L}\mathbf{v}_{k,L+1}^{\top}. \tag{B.3}$$

Now, we can see the term in $Q_{k,[1:L+N]}^{\mathrm{c}}$ whcih acts across the $L$-site and the $L+1$-site is $\mathbf{u}_{k,L}\mathbf{v}_{k,L+1}^{\top}$. Consequently, the boundary term $q_k^{\mathrm{B}}$ in system size $L$ for periodic boundary condition is given by $\mathbf{u}_{k,L}\mathbf{v}_{k,1}^{\top}$.

# C  Recursion equation for $\Theta_k$

$\Theta_k$ can be obtained from the following recursion equation:

$$\Theta_{k+1} = \begin{pmatrix} O_{3(k-2),3} & (\Theta_k)^2 \\ O_{3,3} & O_{3,3(k-2)} \end{pmatrix} + E_{k+1}, \tag{C.1}$$

where $E_k$ is a square matrix of order $3(k-2)$, defined by:

$$(E_k)_{a,b} := \begin{cases} e & (b-a=1), \\ O_{3,3} & (\text{otherwise}), \end{cases} \tag{C.2}$$

where $a$ and $b$ ($1 \leq a, b \leq k-2$) indicate the coordinate for the 3 by 3 block matrices in $E_k$.

In the following, we prove (C.1). We first calculate the $(\Theta_k)^2$ is the RHS of (C.1) for $k \geq 3$. Substituting the expression of $\Theta_k$ of (20), we have

$$\left[(\Theta_k)^2\right]_{a,b} = \sum_{n=1}^{k-2} (\Theta_k)_{a,n} (\Theta_k)_{n,b} = \begin{cases} \sum_{n=a+1}^{b-1} (\Theta_k)_{a,n} (\Theta_k)_{n,b} & (b-a>1), \\ O_{3,3} & (\text{otherwise}), \end{cases} \tag{C.3}$$

where $a$ and $b$ ($1 \leq a, b \leq k-2$) indicate the coordinate for the 3 by 3 block matrices in $\Theta_k$. For the case of $b-a \equiv 1$ (mod 2), there do not exist any integers $n$ that satisfy both $n-a \equiv 1$ (mod 2) and $b-n \equiv 1$ (mod 2) simultaneously. Thus one gets:

$$
\begin{aligned}
\left[(\Theta_k)^2\right]_{a,b} &= \begin{cases} \sum_{l=0}^m (\Theta_k)_{a,a+2l+1} (\Theta_k)_{a+2l+1,b} & (\exists m \in \mathbb{N} : b-a = 2m+2), \\ O_{3,3} & (\text{otherwise}), \end{cases} \\
&= \begin{cases} \sum_{l=0}^m C_l C_{m-l} e & (\exists m \in \mathbb{N} : b-a = 2m+2), \\ O_{3,3} & (\text{otherwise}), \end{cases} \\
&= \begin{cases} C_{m+1} e & (\exists m \in \mathbb{N} : b-a = 2m+2), \\ O_{3,3} & (\text{otherwise}). \end{cases}
\end{aligned}
\tag{C.4}
$$

From (C.4), we can see the RHS of (C.1) is equal to the $\Theta_{k+1}$ of (20). Then we have proved eq. (20) satisfies the recursion equation (C.1).

## D    Examples of local charges for $L = 4$

We give the expressions of the local charges for $L = 4$ case using the building blocks.

$$
Q_2 = \boldsymbol{\sigma}_1 \boldsymbol{\sigma}_2^\top + \boldsymbol{\sigma}_2 \boldsymbol{\sigma}_3^\top + \boldsymbol{\sigma}_3 \boldsymbol{\sigma}_4^\top + \boldsymbol{\sigma}_4 \boldsymbol{\sigma}_1^\top,
\tag{D.1}
$$

$$
Q_3 = \boldsymbol{\sigma}_1 M_2 \boldsymbol{\sigma}_3^\top + \boldsymbol{\sigma}_2 M_3 \boldsymbol{\sigma}_4^\top + \boldsymbol{\sigma}_3 M_4 \boldsymbol{\sigma}_1^\top + \boldsymbol{\sigma}_4 M_1 \boldsymbol{\sigma}_2^\top,
\tag{D.2}
$$

$$
\begin{aligned}
Q_4 = &\boldsymbol{\sigma}_1 M_2 M_3 \boldsymbol{\sigma}_4^\top + \boldsymbol{\sigma}_2 M_3 M_4 \boldsymbol{\sigma}_1^\top + \boldsymbol{\sigma}_3 M_4 M_1 \boldsymbol{\sigma}_2^\top + \boldsymbol{\sigma}_4 M_1 M_2 \boldsymbol{\sigma}_3^\top \\
&+ \boldsymbol{\sigma}_1 \boldsymbol{\sigma}_3^\top + \boldsymbol{\sigma}_2 \boldsymbol{\sigma}_4^\top + \boldsymbol{\sigma}_3 \boldsymbol{\sigma}_1^\top + \boldsymbol{\sigma}_4 \boldsymbol{\sigma}_2^\top.
\end{aligned}
\tag{D.3}
$$

We note that nontrivial coefficients do not appear in these examples up to $Q_4$.

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
