# Peer review of "Matrix product operator representations for the local conserved quantities of the Heisenberg chain"

_SciPost Physics Core, doi:SciPost Phys. Core 6, 069 (2023)_

## Round 1 · Referee Report · Anonymous (Referee 1) · 2023-6-16

Strengths

1- Novel idea. 2- Relatively good explanations.

Weaknesses

1- LPU: Least Publishable Unit.

Report

This work deals with the conserved charges of the Heisenberg XXX spin chain. This is a model whose integrability has been well known for many years. Hans Bethe solved in in 1931, and the algebraic part of the integrability was clarified in the early 80's. However, it appears that new results can be found even today! Recently there has been a renewed interest in the conserved charges in this model and its relatives, with new results emerging starting from entirely new ideas.

I strongly support this type of work: to take an old problem, and to attack it with creative new ideas. Also, I think this paper deserves to be published.

However, I don't think it should be published in SciPost Physics. It appears to be an LPU: least publishable unit. Publishing such a thing is OK if the field is very new, then it is natural that many small results are published separately. However, this field is old, well established. Now if someone wants to publish new results in good journals, then these results should have depth and extent.

I have an offer to the authors: Either they publish this in SciPost Core (which I can accept), or they do a bit of more work, and they add at least one new direction, chosen from these questions:
-Similar MPO for periodic boundary conditions, with a periodic MPO, no boundary term.
-The same result for XXZ, but perhaps even XYZ.
-A similar MPO for the open boundary case, without any extra boundary terms.
-Establishing connections to the standard transfer matrix.

I understand that all of these directions can be difficult. But I do not accept this publication in SciPost Physics, unless new results are added.

The choice is with the authors.

Requested changes

Other small comments:
-In the intro it should be explained that the usual transfer matrix is also an MPO. In fact, these MPO's existed even before MPO's were discovered elsewhere!
-Starting from Section 3.1 it should be explained that this MPO is an MPO with open boundaries, and it should be written in the standard MPO formulation, with boundary vectors for the auxiliary space. Actually, it becomes clear from formula (12) what the authors do, but there should be a more direct explanation. It will be enough to take the sandwhich of the operator on (12) with vectors (1 0 0 0 0) and (0 0 0 0 1) to get H^c. So this is very simple, but it should be explained.
-Also, at this point, it should be explained why this formula works. It is very simple, but it is good to add a short explanation at this point.

---

## Round 1 · Referee Report · Anonymous (Referee 2) · 2023-6-29

Strengths

1- Brings a new result - a disctint way to express the conserved charges of exact integrable models 2- The paper is well written

Weaknesses

1- Although correct the addition of the result to the well established area is not
large.

Report

The authors show in a compact from how to express the local charges of the XXX
model and its SU(N) generalization. The construction is interesting, since
the construction exhibit the general structure for distinct models. In general
it is cumbersome to express these charges, using the standard procedure coming form the the transfer matrix derivatives. In this sense the results are
interesting. But, as pointed out by the other referee, the results presented
are not enough for the publication in the SciPost format, as the present submission. Although brings an original result, its addition to the well established
area of exact integrable models is not substantial. Like the other referee I suggest the authors to submit the paper in the SciPost Core.

Suggestion for improvement: An appendix with all the conserved charges, for a small lattice (ex. L=3, or L=4), will be useful for the readers.

---

## Round 1 · Referee Report · Anonymous (Referee 3) · 2023-7-23

Strengths

  • A new approach to local conserved charges of integrable models using MPO
  • Novel connection to combinatorics

Weaknesses

  • The connection between the current approach and the usual transfer matrix approach has not been fully discussed.

Report

Report: The authors present compact expressions for the local conserved charges of the SU(2) and, more generally, SU($N$) invariant Heisenberg chains. Although these conserved charges have been obtained in previous work, they were expressed in terms of the nested product of local operators or using the doubling notation, which may not be simple enough to make further development. The authors tackle this issue in a neat way. The idea is to use the matrix product operator (MPO) approach to express local conserved charges in a concise form. As a result, the authors have constructed the MPO components of each conserved charge explicitly.

The results are certainly original and will be of interest to the readership. However, as the other referees pointed out, they are not substantial enough for publication in SciPost Physics in their current form. Even if the authors transfer the manuscript to SciPost Core, I would suggest the authors address the following issues in the revision.

  1. Connections to the standard transfer matrix As Referee 1 pointed out, the standard transfer matrix itself is an MPO of bond dimension $N$. This should be emphasized in the introduction. I am actually wondering if the authors can obtain the recursive relation between the MPO components $\Gamma^i_k$ ($k=2,3,4,...$) using the boost operator. Naively, the boost operator can also be written in the form of MPO, at the cost of losing the translational invariance. If this is the case, the commutator between the MPO in Eq. (12) and the boost operator can also be written as an MPO. Of course, the subtlety is that the definition of the boost operator is not compatible with the periodic boundary conditions. Nevertheless, I expect that one could still formally use it in the periodic systems, as discussed in Sklyanin's lecture notes [arXiv:hep-th/9211111]. In any case, I suggest the authors consider this issue, as it might make the derivation of $\Gamma^i_k$ more transparent.

  2. References for boost operator The authors might want to cite references for the boost operator in the introduction. According to Sklyanin's review mentioned above, the boost operator was first introduced by Tetelman in [M. G. Tetelman (1982) Sov. Phys. JETP 55(2), 306–310].

  3. Generalized Catalan numbers I have tried to see the precise relation between the generalized Catalan numbers discussed in Refs. [5, 6, 7] and those in the present manuscript, but couldn't. It would be beneficial to the reader if the authors could provide some remarks on how their $C_{k,n}$ is related to $\alpha_{k,l}$ in the papers by Grabowsk and Mathieu.

Another comment: it is quite likely that the identity in Eq. (23) has been known already in combinatorics. I have the impression that $C_{n+m,n}$ can be interpreted as the number of generalized Dyck paths (the ballot number). In fact, I found the following paper: https://www.researchgate.net/publication/282737525_Generalized_Euler-Segner_formulas_of_Catalan_numbers_and_Motzkin_numbers

in which the generating function of $C_{n+m,n}$ is discussed. There is nothing wrong with including the proof of the identity in the manuscript. However, I would suggest that the authors look up several combinatorics textbooks and see if this was known already.

  1. Minor comments
  2. Page 4, the last line: what is $k$ in $i^k$?
  3. Page 7, 3 lines before Eq. (21): $\Gamma^k_i$ should read $\Gamma^i_k$.
  4. The manuscript contains several grammatical errors that should be corrected before resubmission.

Requested changes

See my comments in Report.

---

## Round 2 · Referee Report · Anonymous (Referee 1) · 2023-8-16

Report

The authors answered my requests, the paper can be published now.

---

## Round 2 · Referee Report · Anonymous (Referee 2) · 2023-8-18

Report

I think the authors made the necessary modifications and I suggest its acceptance in the auggested section of Sci Post

---

## Round 2 · Referee Report · Anonymous (Referee 3) · 2023-8-31

Report

I think the authors have addressed most of the comments raised by the referees. However, I am not fully convinced with one of the authors' answers:

$\quad$ As the referee said, $B$ can be expressed as MPO. However, we have not succeeded in obtaining further results on this topic. The difficulty lies in the difference between two MPO: $Q_k B − B Q_k$. Although $Q_k B$ and $B Q_k$ themselves can be written in MPO, we have not come up with how to make the subtraction of two MPO into a single MPO. We leave this problem for future study and have added the comment at the end of Section 3, in the sentence starting with "One may think we can obtain the recursive relation between...".

In fact, one can rewrite $Q_k B − B Q_k$ as \[
{\rm Tr}_a \left(
\begin{array}{cc}
Q_k B & 0 \\
0 & -B Q_k
\end{array}
\right) ,
\] where $a$ denotes a newly introduced auxiliary space. Then $Q_k B − B Q_k$ can be thought of as an MPO with an enlarged auxiliary space. (I know the authors consider MPOs with open boundaries. But just for illustration, I used an MPO in the form of trace. Actually, one can express MPOs like equation (22) in this form by inserting a boundary matrix.) Since this procedure doubles the dimension of the auxiliary space, I am not sure if this is the best way. But perhaps, the authors might want to examine whether this trick helps simplify the derivation.

I would also suggest the authors consider the comments by the new referee (Bernard Nienhuis).

---

## Round 2 · Referee Report · Bernard Nienhuis (Referee 4) · 2023-8-31

Strengths

  1. MPO as a new approach to compute operators.
  2. Well written, with enough explanation, and no unnecessary side issues.

Weaknesses

  1. As a result of the bottom up construction of the expressions, symbols are introduced at a time the reader does not know their purpose. This makes it harder to understand and remember the definitions.
  2. The advance in the subject is less than significant
  3. In my opinion the new expressions for the charges are not simpler than what is offered in the literature. They are just simpler.

Report

This paper gives an explicit expression for the local conserved quantities of the (SU(2)) Heisenberg chain, and its SU(N) generalizations.
The subject of the explicit form of conserved quantities of integrable models has received more attention recently. This may well be useful, as these explicit forms may be exploitable in the Generalized Gibbs Ensemble, and in Generalized Thermodynamics, subjects in which the phrase generalized refers to the inclusion of more conserved quantities than just energy and momentum. The result itself presented in the paper is not new, and one of the sections of the paper is devoted to describing the known results.
The motivation for presenting a known result in a new form is twofold. One is that the expressions in the literature are relatively complicated, and a more simple form would be desirable. The other is that the technique of tensor networks is used with increasing frequency for calculations on quantum systems, so that it might be useful to bring the conserved charges in that form.
The new form in which the conserved quantities are presented is a called a matrix product operator (MPO). It is a product of N matrices used for a chain of N spins. The matrix elements are operators that each act in the Hilbert space of one of the spins. This is indeed a form of a tensor network. But is should not be confused with a matrix product state, which is an entirely different concept.

I fully agree that it may be useful to write the old results in a new form to make them amenable to new techiques and methods. But to say that the MPO representation is simpler than the old, known form is too much. It certainly requires more explanation. The matrices are set up such that of the whole matrix product one element is the desired result, and it should still be supplemented with the terms that involve the spins near one end of the chain and also near the other end.

I find the paper well written. In my opinion the paper certainly deserves publication. I find the result of the paper a significant achievement. Its usefulness is yet to be demonstrated. The expectations for SciPost Physics Core are (in my opinion) not quite met: 1) It can be argued that the rewriting of the local charges is an important problem, to make them accessible for applications. It has been addressed with appropriate methods and above average originality.
However, 2) in my opinion it does not meet the expectation of significantly advancing knowledge or underanding of the field.

Requested changes

  1. typo: page 9 line 6 the symbol x in the expression for B should be i.

  2. The word component is used for the element of a vector or of a matrix, but also, like in the line before eq.(3) in the general (and vaguer) meaning of "building block". The use of the definite article (THE components) suggests that the precise meaning of the word is obvious. It is not to me. I would say that omission of the definite article "the" is better, or e.g. replacement of the word "components" by "terms".

---

## Round 2 · Author Response

Author comments upon resubmission

We are very grateful to the referees for the careful review of our manuscript and also grateful to the editors for their arrangement for the peer review of our manuscript.

We made a couple of changes based on the comment from the referees, and please find our replies to Referee 1, Referee 2, and Referee 3 below.

Reply to Referee 1

We are grateful to Referee 1 for his/her positive evaluation and strong support of our work. We also understand and follow the referee's suggestion to transfer our work to SciPost Physics Core.

The reply to the referee's request is the following:

-In the intro it should be explained that the usual transfer matrix is also an MPO. In fact, these MPO's existed even before MPO's were discovered elsewhere!

We have added an explanation about the relationship between the transfer matrix and the MPO at the end of the paragraph starting with "Over the past two decades..." on page 1. After the sentence "However, despite the transfer matrix, which is the source of $Q_k$, being defined by the MPO constructed from the Lax operator, ... ", we have stated, "For example, the transfer matrix of the $\mathrm{SU(N)}$ invariant chain is the MPO with bond dimension $N$. It is worth noting that the fact that the transfer matrix is expressed as an MPO had been known even before the term "MPO" was coined."

-Starting from Section 3.1 it should be explained that this MPO is an MPO with open boundaries, and it should be written in the standard MPO formulation, with boundary vectors for the auxiliary space. Actually, it becomes clear from formula (12) what the authors do, but there should be a more direct explanation. It will be enough to take the sandwhich of the operator on (12) with vectors (1 0 0 0 0) and (0 0 0 0 1) to get H^c. So this is very simple, but it should be explained.

We have added an explanation that our MPO is an open boundary MPO at the end of the first paragraph of section 3 in the sentence, "We show that the local conserved quantities can be represented by an open boundary MPO constructed with the upper triangular matrix $\Gamma_k^i$." We also add the boundary vector forms for the bulk below Eq.(12) and in Eq.(22).

-Also, at this point, it should be explained why this formula works. It is very simple, but it is good to add a short explanation at this point.

The important thing for our formula to work is that the components of the local charges of XXX chain have the form of the nested product, as seen in Eq.(2) and Eq.(8), and this is not the case for the anisotropic generalization. We have added the explanation on this point in the paragraph at the end of Section 3, in the sentence starting with "We briefly explain why the local charges can be represented simply...".

Reply to Referee 2

We thank the referee for his/her positive comments and helpful suggestions. We also understand and follow the referee's suggestion to transfer our work to SciPost Physics Core. Following the suggestion of Referee 2, we added the appendix with all the conserved charges for L=4.

Reply to Referee 3

We thank the referee for his/her very careful reading of our paper and positive comments. We also thank the referee for pointing out several typographical errors. The reply to the referee's request is the following:

-1. Connections to the standard transfer matrix
As Referee 1 pointed out, the standard transfer matrix itself is an MPO of bond dimension N. This should be emphasized in the introduction. I am actually wondering if the authors can obtain the recursive relation between the MPO components $\Gamma_k^i (k=2,3,4,\dots)$ using the boost operator. Naively, the boost operator can also be written in the form of MPO, at the cost of losing the translational invariance. If this is the case, the commutator between the MPO in Eq. (12) and the boost operator can also be written as an MPO. Of course, the subtlety is that the definition of the boost operator is not compatible with the periodic boundary conditions. Nevertheless, I expect that one could still formally use it in the periodic systems, as discussed in Sklyanin's lecture notes [arXiv:hep-th/9211111]. In any case, I suggest the authors consider this issue, as it might make the derivation of $\Gamma_k^i$ more transparent.

-2. References for boost operator
The authors might want to cite references for the boost operator in the introduction. According to Sklyanin's review mentioned above, the boost operator was first introduced by Tetelman in [M. G. Tetelman (1982) Sov. Phys. JETP 55(2), 306–310].

We have attempted to obtain the recursive relation for $\Gamma_k^i$ using the Boost operator $B$: $[Q_k, B]=Q_{k+1}$. As the referee said, $B$ can be expressed as MPO. However, we have not succeeded in obtaining further results on this topic. The difficulty lies in the difference between two MPO: $Q_k B -B Q_k$. Although $Q_k B$ and $BQ_k$ themselves can be written in MPO, we have not come up with how to make the subtraction of two MPO into a single MPO. We leave this problem for future study and have added the comment at the end of Section 3, in the sentence starting with "One may think we can obtain the recursive relation between...".

-3. Generalized Catalan numbers
I have tried to see the precise relation between the generalized Catalan numbers discussed in Refs. [5, 6, 7] and those in the present manuscript, but couldn't. It would be beneficial to the reader if the authors could provide some remarks on how their $C_{n+m,n}$ is related to $\alpha_{k,l}$ in the papers by Grabowsk and Mathieu.

We have added the explanation for the translation between $C_{n+m,n}$ and $\alpha_{k,l}$ in the footnote of page 4.

-Another comment: it is quite likely that the identity in Eq. (23) has been known already in combinatorics. I have the impression that $C_{n+m,n}$ can be interpreted as the number of generalized Dyck paths (the ballot number). In fact, I found the following paper: https://www.researchgate.net/publication/282737525_Generalized_Euler-Segner_formulas_of_Catalan_numbers_and_Motzkin_numbers
in which the generating function of $C_{n+m,n}$ is discussed. There is nothing wrong with including the proof of the identity in the manuscript. However, I would suggest that the authors look up several combinatorics textbooks and see if this was known already.

We have referred to some combinatorics textbooks (Concrete Mathematics[Graham, Knuth, Patashnik], Introduction to Combinatorial Analysis[Riordan]), but we did not find the statement about the formula (24). We find a similar but not the same formula investigated in [Shapiro, Louis W. "A Catalan triangle." Discrete Mathematics 14.1 (1976): 83-90]. We cite this thesis and the paper the referee mentioned in the main manuscript and add a comment starting with "After the completion of this work, ..." at the end of subsection 4.1.

---

## Round 2 · List of Changes

• Added a remark on the fact that the transfer matrix is also MPO in the introduction.
  • Added an explanation about the relationships between the generalized Catalan numbers and the coefficient given by Grabowski and Mathieu in subsection 2.1.
  • Added a remark on the fact that our MPO is an MPO with open boundaries and how to get the bulk term of the local conserved quantities from our MPO using boundary vectors around eq.(22) in Section 3.
  • Added a brief explanation of why our MPO formula for $Q_k$ works at the end of Section 3.
  • Added a comment on the boost operator at the end of Section 3.
  • Added some references about the identity of the generalized Catalan numbers in subsection 4.1.
  • Give an explicit form of $Q_k$($k = 2, 3, 4$) for $L = 4$ in Appendix D.
  • Some typos and grammatical errors have been fixed.

---

## Round 3 · Author Response

We are very grateful to the referees for the careful review of our manuscript and also grateful to the editors for their arrangement for the peer review of our manuscript.

We made some changes based on the comments from the referees, and please find our replies to individual reports below.

Reply to Report 1 and Report 2

We express our gratitude to the referees for their recommendations regarding the publication of our paper.

Reply to Report 3 by Bernard Nienhuis

We express our gratitude to Prof. Nienhuis for his careful reading of our paper, and we are also grateful for pointing out the terminological ambiguities. We improved the manuscript following your requested changes. The reply to other comments is the following:

I fully agree that it may be useful to write the old results in a new form to make them amenable to new techiques and methods. But to say that the MPO representation is simpler than the old, known form is too much. It certainly requires more explanation.

As you noted, the explanation for the simplicity of our MPO when compared to the old known result was ambiguous. We would like to emphasize the simplicity of the coefficients appearing in the MPO. Our MPO is constructed from only the usual Catalan number. In contrast, the previously known formula for the charges, eq(4) in our manuscript, requires the generalized Catalan number. Finding the regularity of the coefficients in the local charges is generally a challenging task (please see also our reply below). Thus, we believe it is worth mentioning that the coefficients become simple in MPO representation.

We emphasized the point that coefficients are simplified in our MPO representation in the paragraph around the end of Subsection 3.3, starting with "We note that $\Gamma^{i}_{k}$ is only involved with the usual Catalan number,...", and in the sentence in the first paragraph of Section 5, starting with "Especially the coefficients appearing in the MPO are simpler ...".

The matrices are set up such that of the whole matrix product one element is the desired result, and it should still be supplemented with the terms that involve the spins near one end of the chain and also near the other end.

Thank you for pointing out the concerns related to boundary treatment, which makes the expression cumbersome. However, in the infinite chain case, the boundary terms in the local charges become irrelevant, and it is enough to consider only the bulk term.

We have added the explanation about the case of the infinite chain in the middle of subsection 3.3, in the new paragraph starting with "For the case that the Hamiltonian is defined on the infinite chain, ...".

I find the paper well written. In my opinion the paper certainly deserves publication. I find the result of the paper a significant achievement. Its usefulness is yet to be demonstrated. The expectations for SciPost Physics Core are (in my opinion) not quite met: 1) It can be argued that the rewriting of the local charges is an important problem, to make them accessible for applications. It has been addressed with appropriate methods and above average originality.
However, 2) in my opinion it does not meet the expectation of significantly advancing knowledge or underanding of the field.

Thank you for recognizing the significance of our result and considering our paper worthy of publication.

One of the advantages of our MPO representation is that it may give a new way to predict the regularity of the local charges.

For almost all interacting integrable systems, the coefficients appearing in the local charges are so complicated that we cannot infer their regularities. For example, the explicit expressions for the local charges in the XXZ chain had been a mystery for about thirty years since the progress of the isotropic XXX cases. This problem was recently solved utilizing the Temperley-Lieb algebra in your seminal paper[Nienhuis, Huijgen, 2021, J. Phys. A: Math]. We think that, in terms of the Temperley-Lieb algebra, the regularity for the coefficients becomes more straightforward than their bare expression in the spin operator, and this simplification follows the discovery of the general expressions for the charges.

In this spirit, to predict the regularity of local charges, it may be helpful to develop an alternative way to express local charges. In fact, in our case, the coefficients have been simplified, as noted in the above reply. Although, in this paper, we treat the XXX chain where the structure of the local charges has been well understood, we believe our strategy will offer a new way to analyze the regularity of unknown charges. Detailed investigation on this topic goes beyond the current paper, and we left this study as a future problem, which is stated in the summary, in the paragraph starting with "Our strategy may be useful to find the general expressions...".

Reply to Report 4

We express our gratitude to the referee for his/her careful review of our previous resubmission and helpful suggestions.

The reply to the referee's request is the following:

In fact, one can rewrite $Q_k B − B Q_k$ as

$$ \mathrm{Tr}_a \begin{pmatrix}Q_k B & O \ O & −BQ_k\end{pmatrix}, $$
where $a$ denotes a newly introduced auxiliary space. Then $Q_kB−BQ_k$ can be thought of as an MPO with an enlarged auxiliary space. (I know the authors consider MPOs with open boundaries. But just for illustration, I used an MPO in the form of trace. Actually, one can express MPOs like equation (22) in this form by inserting a boundary matrix.) Since this procedure doubles the dimension of the auxiliary space, I am not sure if this is the best way. But perhaps, the authors might want to examine whether this trick helps simplify the derivation.

Thank you for the insight into the relation between the boost operation and MPO. We followed the method presented by the referee to obtain the MPO representation recursively using the boost operator. For simplicity, we consider the infinite chain case where the boundary effect becomes irrelevant. We added the explanation in the new subsection 3.4, starting with "One may think we can obtain the recursive relation ...". We also noted that boost-derived charges are slightly different from our $Q_k$ in the linear combination of the lower-order charges, and subsequently, the expressions of the MPO are also different.

---

## Round 3 · List of Changes

• Added a remark on the MPO representation of the local charges of infinite chains in Subsection 3.3.
  • Added a new subsection (Subsection 3.4) to explain how to construct the MPO recursively with the boost operation.
  • Added an explanation about the advantage of our MPO representation of the local charges in Section 5 and Subsection 3.3.
  • Some typos and confusing expressions have been fixed.

---

## Editorial Decision

published